# The secondary messenger ppGpp interferes with cAMP-CRP regulon by promoting CRP acetylation in *Escherichia coli*

**Chunghwan Ro, Michael Cashel, Llorenç Fernández-Coll** *

*Eunice Kennedy Shriver* National Institute of Child Health and Development, NIH, Bethesda, Maryland, United States of America

* llfernandezcoll@gmail.com

**Data Availability Statement:** All relevant data are within the manuscript and its Supporting Information files.

**Funding:** This research was funded by the intramural research program of the Eunice

## Abstract

The cAMP-CRP regulon coordinates transcription regulation of several energy-related genes, the *lac* operon among them. Lactose, or IPTG, induces the *lac* operon expression by binding to the LacI repressor, and releasing it from the promoter sequence. At the same time, the expression of the *lac* operon requires the presence of the CRP-cAMP complex, which promotes the binding of the RNA polymerase to the promoter region. The modified nucleotide cAMP accumulates in the absence of glucose and binds to the CRP protein, but its ability to bind to DNA can be impaired by lysine-acetylation of CRP. Here we add another layer of control, as acetylation of CRP seems to be modified by ppGpp. In cells grown in glycerol minimal media, ppGpp seems to repress the expression of *lacZ*, where Δ*relA* mutants show higher expression of *lacZ* than in WT. These differences between the WT and Δ*relA* strains seem to depend on the levels of acetylated CRP. During the growth in minimal media supplemented with glycerol, ppGpp promotes the acetylation of CRP by the Nε-lysine acetyltransferases YfiQ. Moreover, the expression of the different genes involved in the production and degradation of Acetyl-phosphate (*ackA-pta*) and the enzymatic acetylation of proteins (*yfiQ*) are stimulated by the presence of ppGpp, depending on the growth conditions.

## Introduction

The second messengers (p)ppGpp (Guanosine pentaphosphate and Guanosine tetraphosphate) have long been known to accumulate and to alter gene expression as a response to nutritional and physical stress throughout the bacterial kingdom [1, 2]. Increased levels of ppGpp will stop bacterial growth, inhibiting the production of RNA, DNA and proteins [3–5]. Maintaining the balance of ppGpp is crucial; too much inhibits growth and too little makes the cell more vulnerable to nutritional stress [6].

In *Escherichia coli*, the levels of ppGpp depend on the balance between synthesis activity of RelA and the bifunctional activity of SpoT, which contains a weak synthetase and a hydrolase domain. The synthetase of RelA is activated by binding to ribosomes and sensing amino acid

Kennedy Shriver National Institute of Child Health and Human Development (NICHD), NIH. CR recieved a stipend from the NIH Office of Intramural Training & Education. Funders had no role in the design of the study; in the collection, analyses, or interpretation of data; in the writing of the manuscript, or in the decision to publish the results.

**Competing interests:** The authors have declared that no competing interests exist.

starvation when uncharged tRNA binds to ribosomal A sites [7–9]. In contrast, SpoT-dependent accumulation of ppGpp is promoted by carbon source, fatty acid, iron, nitrogen, phosphate starvation, pH and oxidative stress [10–16]. The balance between synthesis and hydrolysis in SpoT can be tilted towards one or the other through the binding of different proteins to SpoT, such as acyl carrier protein, Rsd or YtfK [17–19].

In Gram-negative bacteria, ppGpp regulates transcription initiation by binding directly to RNA polymerase at two sites [3, 20, 21]. The discriminator sequences, located between the promoter -10 and +1 regions, determines whether ppGpp stimulates or inhibits transcription, depending on whether it is AT-rich or GC-rich respectively [20, 22, 23]. Alternatively, ppGpp can directly bind to proteins to alter their catalytic activities [24–26].

The secondary messenger ppGpp is also known to be important for the proper regulation of genes during diauxic growth [27, 28]. This phenomenon, discovered by Monod in 1947 [29], occurs when bacteria are exposed to limiting amounts of glucose and an excess of a less efficient carbohydrate (such as lactose). Under these circumstances bacteria will utilize exclusively glucose until it runs out. Then, cells will stop growing (diauxic lag time) and they alter their gene expression pattern to allow lactose-fueled growth. The level of ppGpp reaches a peak during the diauxic lag time, before resuming growth utilizing lactose [28]. Although several master regulators control diauxic shift [30, 31], ppGpp affects the length of the diauxic lag time by increasing the levels of acetyl-phosphate and, consequently, the levels of acetylated proteins [28].

In *Escherichia coli*, the phosphotransferase system (PTS) plays a major role in the control of diauxic growth. The preferential glucose uptake is accompanied by its phosphorylation to glucose-6P by the PTS, using phosphoenolpyruvate (PEP) as a phosphate donor. The PTS components will oscillate between their phosphorylated and non-phosphorylated forms. The presence of glucose in the media will promote a continued formation of glucose-6P that will favor the conversion of the PTS components to their non-phosphorylated forms. The non-phosphorylated form of EIIAglc (one of the components of the PTS) binds and inhibits the transport machinery for non-PTS sugars (such as lactose), while inhibiting the synthesis of cAMP. In the absence of glucose, the PTS components remain phosphorylated and can activate the adenylate cyclase synthesis of cAMP [30, 31]. The cAMP receptor protein (CRP) complexed with cAMP, plays a key role in gene expression in *Escherichia coli*. The cAMP-CRP complex binds to specific sites within the promoter regions of several operons, stimulating or inhibiting transcription initiation [32, 33]. When bacteria grow at a constant temperature without starvation, all cellular components are synthesized exponentially. When the balanced growth rates are determined not by limiting nutrient abundance but by the ability of the cell to use different nutrients in excess, a correlation of the amount of protein, DNA and RNA with the bacterial growth rate is observed [34]. As previously mentioned, this correlation depends on the presence of ppGpp [4, 5]. Using similar conditions, an inverse correlation was found between the levels of (p)ppGpp and growth rate, where higher levels of (p)ppGpp correlate with lower growth rates [35]. In a similar fashion, the expression of carbon catabolic genes (such as *lacZ*) is inversely proportional to growth rates upon limitation of carbon sources [36]. The levels of the catabolic growth seem to correlate with the levels of cAMP. By controlling the influx of carbon sources, cAMP controls the synthesis of precursors of amino acids that would otherwise control the bacteria proteome [36]. It is interesting to note that limiting the amount of amino acid precursors will promote the RelA-dependent synthesis of ppGpp [28, 37].

As previously mentioned, increasing the levels of ppGpp will negatively affect the growth rate [3]. A recent study [38] has shown that decreasing bacterial growth rate by artificially increasing the levels of ppGpp (by overexpressing a truncated RelA protein with constitutive ppGpp synthesis) results in increasing levels of catabolic enzymes. The study

shows a similar inverse correlation between growth rate and expression of carbon catabolic genes as observed when the growth rate is determined by using different carbohydrates, that is attributed to cAMP [36]. A question arises as to whether there is a crosstalk between both secondary messengers, ppGpp and cAMP. Here we use the catabolic gene *lacZ* as a reporter of the effects that ppGpp may exert over the cAMP-CRP complex. We explore the possibility that ppGpp controls the ability of CRP to bind to the promoter region. Considering that CRP can be acetylated by acetyl-phosphate, changing its ability to bind DNA [39], and that ppGpp controls the levels of acetyl-phosphate [28], we suggest that ppGpp may alter the acetylation state of CRP.

## Materials and methods

### Strain construction and growth media

All strains and plasmids used are listed in S1 Table. The different strains were constructed by P1 transduction of gene deletions that came from the Keio collection (marked with kanamycin resistance cassette) or from BW16470 (Δ*ackA* Δ*pta zej223*::Tn10). M9 minimal media was prepared as previously described [4], supplemented with either glucose 0.2%, 0.4% lactose, 0.4% glycerol or 0.2% of N-acetylglucosamine (NAG), when indicated. The following antibiotics were added at the following concentrations: 15 μg/ml tetracycline (Tc) and 20 μg/ml kanamycin (Km). IPTG (Isopropyl-β-D-thio-galactopyranoside) was added to a final concentration of 1 mM.

### β-galactosidase assays

Two independent cultures (25 ml in 125 ml flasks) for each strain were grown in minimal media with aeration at 37˚C to an $OD_{600}$ of 0.1, supplemented with IPTG 1 mM. 1 ml samples were harvested, placed in ice and β-galactosidase activity determined as previously described in [40]; using ortho-nitrophenyl-β-galactoside (ONPG) as a substrate, and the Miller Units are calculated as $(1000 \times (OD_{420} - 1.75 \times OD_{550})) / (time \times volume \times OD_{600})$.

### Western blots

Cells were grown by duplicate in minimal media with aeration at 37˚C up to an $OD_{600}$ of 0.1. Then,1 ml culture was centrifuged, and 1X Sample Buffer (Thermo Fisher) was added to the cell pellet. Samples were boiled and resolved on a NuPAGE 10% Bis-Tris Gel in MES SDS Running Buffer from Thermo Fisher. Proteins were transferred onto a nitrocellulose membrane with an iBlot gel transfer system. Crp and RpoA were detected with α-Crp, α-RpoA mouse antibody (Biolegend) diluted 1/5000 in PBS-T with 2.5% of Milk. To detect lysine-acetylated proteins mouse anti-AcK antibody (Cell Signaling Technology) diluted 1/1000 in TBS-T with 5% of BSA was used. IRDye 800CW Donkey anti-mouse (Li-Cor) was used as secondary antibody. The signal was detected with Odyssey CLX System from Li-Cor, and the intensity of the different bands was quantified using the software Image Studio Lite from Li-Cor.

### cAMP quantification

The levels of cAMP (Cyclic adenosine monophosphate) were determined through the usage of cAMP-Glo™ Max Assay kit (Promega, cat. #V1681). Briefly, cells were grown by duplicate in minimal media (50 ml of media in 250 ml flasks) with aeration at 37˚C to an $OD_{600}$ of 0.1. 50 ml were harvested by centrifugation and resuspended with Induction Buffer provided in the kit. Samples were then processed as indicated by the manufacturer. This commercial kit is based on the principle that cAMP stimulates protein kinase A activity; this stimulation depletes

ATP in the reaction resulting in decreased light production in a coupled luciferase reaction. Luminescence was then detected with a Synergy HT plate reader and the amount of cAMP adjusted by the culture's $A_{600}$.

## ppGpp measurement

The levels of ppGpp were measured by Thin Layer Chromatography (TLC) after labelling with P32 as previously described in [41].

## Diauxic shift measurement

Cells were grown in M9 minimal media supplemented with 0.025% of glucose and 0.4% of lactose, in a 96-well plate at 37˚C with aeration. $OD_{600}$ was measured every 10 minutes for a total of 12 hours with a Synergy HT plate reader. The Diauxic lag time was determined as described in [28].

## Gene expression by RT-qPCR

Expression of *cyaA*, *pta*, *ackA*, *yfiQ* and *cobB* were determined by RT-qPCR. Briefly, 2 independent cultures for each strain were grown up to $OD_{600}$ 0.1, when samples were subject to RNA isolation with Trizol (ThermoFisher) as indicated by the manufacturer. The RNA samples were retrotranscribed into cDNA using the high-capacity cDNA reverse transcription kit from Applied Biosystems and target genes were amplified with SYBR Green PCR Master Mix from Applied Biosystems in a LightCycler 480 instrument. The relative gene expression was determined with the comparative CT method, as described in [42] with 3 technical replicates (6 values for each experimental condition). The *scpA* gene was used as an internal control, since it is not affected by either ppGpp or CRP [27], compared to other house-keeping genes commonly used (such as rRNA operons, *gyrA*, *parC*, *zwf* or *gapA*). Primers used in this study are listed in S2 Table.

# Results

## ppGpp interferes with the CRP-cAMP regulon

Transcription initiation from the *lacZ* promoter is negatively regulated by LacI and is positively regulated by CRP-cAMP. As previously mentioned, the expression of carbon *lacZ* is inversely proportional to growth rates, when the growth rates are determined by using different carbohydrates [36]. In this study, You et al. used 1 mM IPTG to ensure that LacI will not bind to the *lacZ* promoter, giving a similar behavior as using a LacI-deficient background. As a consequence, the expression of *lacZ* is exclusively dependent on CRP-cAMP, which becomes a reporter for this complex [36]. Considering the role of ppGpp controlling the growth rate [3–5], we wanted to determine if ppGpp has any effect over this correlation.

The strain MG1655 (WT) was grown in M9 minimal media with different carbon sources up to exponential phase, and then samples were taken to determine *lacZ* expression (by measuring β-galactosidase activity). The ability to use each carbon source determines the growth rate of each sample. Then, the *lacZ* expression was correlated to the growth rate (Fig 1A). As previously described [36], the expression of *lacZ* seems to inversely correlate with the growth rate, where slower growth rates show higher β-galactosidase activity.

Strains deficient in ppGpp (ΔrelA ΔspoT) do not grow in minimal media without amino acids [10], therefore we used a strain lacking the RelA synthetase. RelA-dependent synthesis of ppGpp responds to starvation of amino acids [43], but also it can sense carbon source starvation in the absence of amino acids [28, 44]. Therefore, under the tested conditions, most of

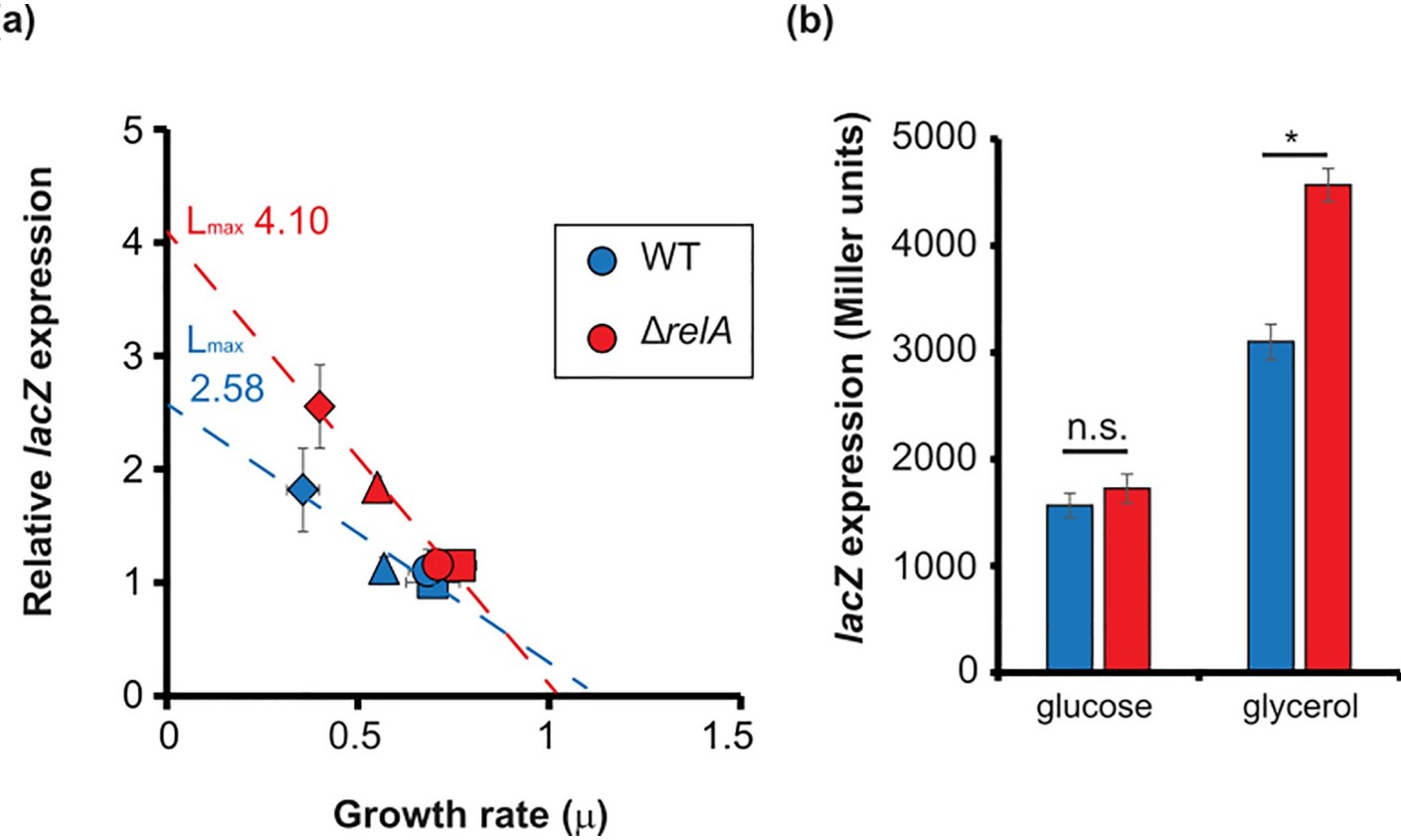

**Fig 1. ppGpp affects *lacZ* expression dependent on growth rate.** MG1655 (WT) and CF18005 (Δ*relA*) were grown up to exponential phase (OD$_{600}$ of 0.1) in M9 minimal media supplemented with glucose (squares), glycerol (diamonds), lactose (triangles) and NAG (circles). The *lacZ* expression was measured from 2 biological replicates and 3 technical replicates. The levels of *lacZ* expression levels were normalized to the expression of WT strain in M9 with glucose. Error bars represent SD. Lmax was calculated as in [36].

the ppGpp will be synthesized by RelA. We have measured the levels of ppGpp in minimal media supplemented with either glucose or glycerol, and we observe, as expected, that most of the ppGpp significantly decreases in the absence of *relA* (S1 Fig). The small amounts of ppGpp detected in absence of RelA, is produced by SpoT, which is sufficient to keep the cells alive under this conditions. As previously mentioned, the levels of ppGpp inversely correlates with the growth rate [35]. Thus, we observe higher levels of ppGpp in glycerol than in glucose in the WT strain (S1 Fig).

As observed for WT, the *lacZ* expression in Δ*relA* strain is inversely proportional to the growth rate, but with a different slope than WT (Fig 1A). We observe that cells growing on M9 minimal media supplemented with glucose (Fig 1A, squares), display the fastest growth rate but they show no differences between WT and Δ*relA*. However, with cells growing slower in glycerol (Fig 1A, diamonds), the Δ*relA* strain shows significantly higher *lacZ* expression than WT (Fig 1A). The absolute values of β-galactosidase activity from the WT and Δ*relA* strains grown in glucose and glycerol can be found in Fig 1B. The maximum beta-galactosidase activity (Lmax, intersection with X-axis as described in [36]) is the highest (theoretical) β-galactosidase activity that these strains can reach. The Lmax was measured for each strain and added to Fig 1A. While the WT strains can reach maximum β-galactosidase activity of 2.58 times the activity in glucose, the Δ*relA* mutants can reach a maximum β-galactosidase activity of 4.10 times the activity of WT in glucose.

The first question is whether ppGpp can affect the expression of *lacZ* by directly affecting the promoter expression or indirectly by affecting the CRP-cAMP complex. As previously mentioned, genes directly regulated by ppGpp show a GC-rich discriminator (region between -10 box and +1) if repressed by ppGpp or an AT-rich discriminator if stimulated [20]. Neither lacUV5 nor *lac* promoter show such discriminator, suggesting that the any effect is indirect. Moreover, *In vitro* experiments have previously shown that ppGpp affects the expression of the *lac* operon but only in the presence of the CRP binding site [45, 46], suggesting that any effect of ppGpp over the *lac* operon requires CRP-cAMP. As corroboration, we determined the expression of the promoter lacUV5, (using a multicopy plasmid with a lacUV5 promoter, which lacks the CRP binding site, fused to RFP-coding gene) in the presence or absence of *relA*. As previously described *in vitro* [46], the *lac* operon does not respond to the presence or absence of *relA* in absence of the CRP binding site (S2 Fig). These experiments discard the possibility of ppGpp directly affect the *lac* promoter. Instead, ppGpp indirectly affects the expression of *lacZ* by affecting CRP or cAMP.

## The effect of ppGpp over *lacZ* depends on acetylation of CRP

We think that ppGpp may affect the CRP-cAMP regulon in 3 possible ways: 1) affecting the levels of cAMP, 2) affecting the levels of CRP or 3) affecting the activity of CRP. Then, the levels of CRP and cAMP were determined under the same conditions as before, in minimal media supplemented with glucose or glycerol (S3 Fig). Despite reports that (p)ppGpp inhibits promoter 2 of *crp* under stress conditions [47], we find no difference in the levels of CRP between WT and Δ*relA* (S3A Fig). At the same time, no significant differences were observed in the levels of cAMP between WT and Δ*relA* (S3B Fig), nor in the expression of the adenylate cyclase (*cyaA*) responsible for producing cAMP (S4 Fig).

Then, since ppGpp does not affect the levels of CRP and cAMP under our conditions, we decided to determine whether ppGpp was affecting the activity of CRP. As previously mentioned, increased ppGpp is associated with increased acetyl-phosphate [28] and the activity of CRP can be modified by its acetylation with acetyl-phosphate [39]; therefore, it seemed possible that ppGpp can affect the acetylation of CRP.

As before, we measured *lacZ* expression by measuring β-galactosidase activity, but this time using a double mutant deleting the *ackA-pta* operon, responsible for synthesis and degradation of acetyl-phosphate. We grew WT and Δ*relA* with or without the *ackA-pta* operon in M9 minimal media supplemented with glycerol, and the β-galactosidase activity was measured (Fig 2A). In the presence of the *ackA-pta* operon, the mutant Δ*relA* showed increased *lacZ* expression compared to WT, as before (Fig 1). However, in absence of *ackA-pta*, there is no difference in the *lacZ* expression between the strains with or without *relA*, suggesting that the differences in *lacZ* expression produced by ppGpp depend on the presence of acetyl-phosphate and the levels of acetylated proteins.

Apart from the non-enzymatic acetylation of proteins produced by acetyl-phosphate, proteins can be acetylated by Nε-lysine acetyltransferases (KATs). In *Escherichia coli*, the main KAT is YfiQ (also known as PatZ or Pka), although other KATs exist [48]. Instead, CobB is responsible for deacetylation of proteins, independently of the method of acetylation [49]. Then, we decided to also determine the *lacZ* expression in presence or absence of *yfiQ* or *cobB* (Fig 2B). In absence of YfiQ, the expression differences between WT and Δ*relA* strains also disappear (Fig 2B), as seen in Δ*ackA-pta* mutants (Fig 2A), suggesting that both enzymatic and non-enzymatic acetylation may be involved. In absence of CobB, we observe a general decrease on the levels of *lacZ* expression in both strains (Fig 2B), but with a larger difference between WT and Δ*relA*. In absence of CobB, the amount of acetylated CRP is predicted to increase,

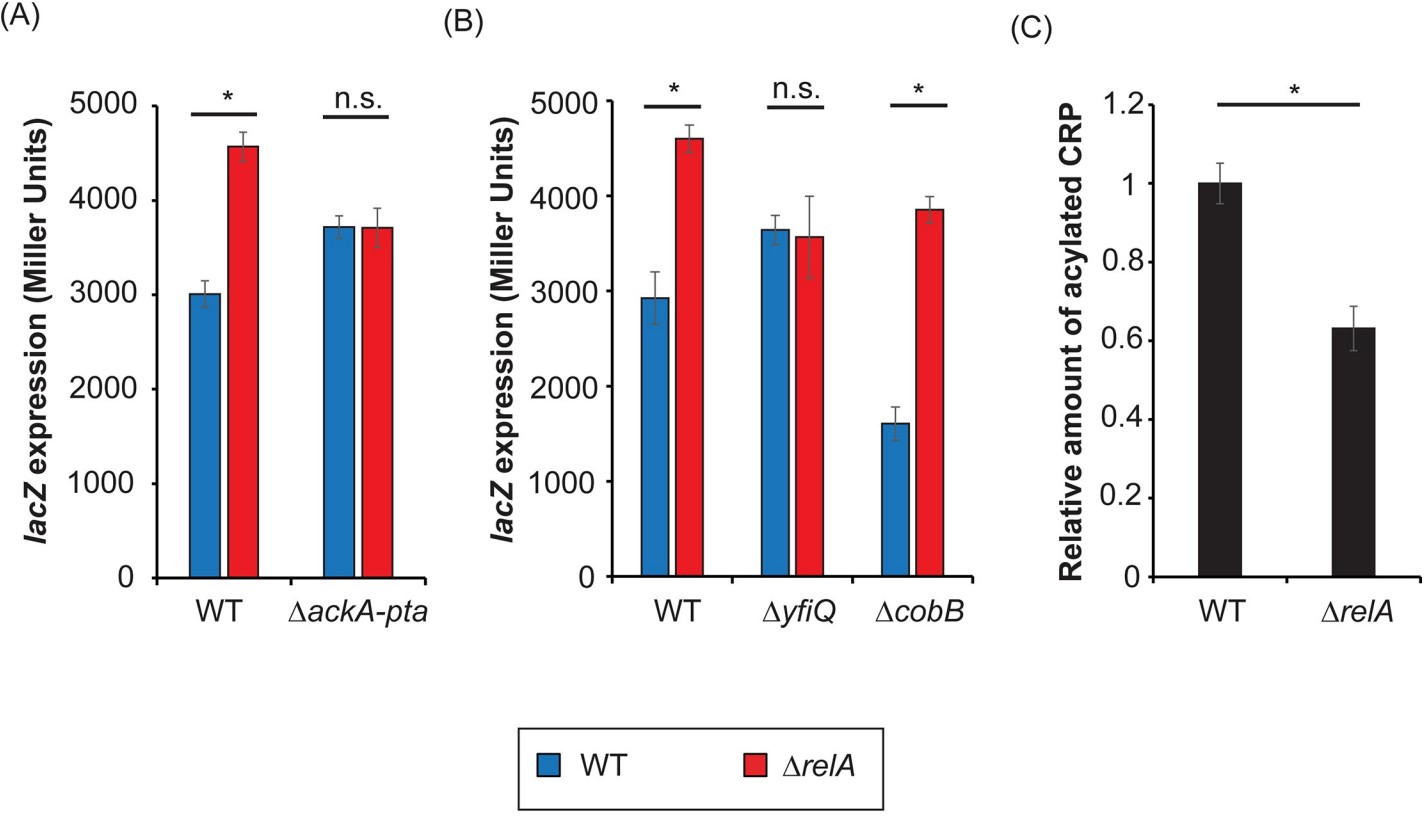

**Fig 2. ppGpp affects *lacZ* expression by promoting CRP acetylation.** (A) β-galactosidase activity measure from samples of MG1655 (WT) and CF18005 (Δ*relA*) and their isogenic Δ*ackA-pta* mutants grown up to exponential phase (OD$_{600nm}$ 0.1) in M9 minimal media supplemented with glycerol 0.4%. Error bars show standard deviation of 2 biological samples and 3 technical replicates. (B) As before, WT and Δ*relA* strains, together with their isogenic Δ*yfiQ* and Δ*cobB* mutants were grown in M9 minimal media supplemented with glycerol 0.4% up to exponential phase (OD$_{600nm}$ 0.1) and β-galactosidase activity was measured. Error bars show standard deviation of 2 biological samples and 3 technical replicates. (C) The amount of acetylated CRP, relative to the WT stain, was measured by Western blot from total extracts of WT and Δ*relA* strains grown in M9 minimal media supplemented with glycerol 0.4% and grown up to exponential phase (OD$_{600nm}$ 0.1). Western blots and quantification are presented in S6 Fig. Error bars show standard deviation of 2 biological samples. Statistical significance was measured with T-student test (*p-value < 0.05, n.s. p-value > 0.05).

accounting for the general decrease on *lacZ* expression, and presumably there would be more acetylated CRP in WT strains than Δ*relA* strains.

As previously mentioned, ppGpp affects the length of the diauxic lag time by changing the levels of acetyl-phosphate [28]. There, we observed that Δ*relA* mutants had longer diauxic lag times than WT in the presence of the *ackA-pta* operon, but they showed similar diauxic lag times in its absence. Considering that YfiQ seem to be required for the effect of ppGpp over the *lacZ* expression (Fig 2B), we wanted to determine if YfiQ is necessary for the effects of ppGpp on the diauxic lag time (S5 Fig). No difference was observed in the diauxic lag time in presence or absence of YfiQ (S5 Fig).

Then, we decided to determine the relative amount of acetylated CRP by Western blot in the presence or the absence of RelA (Fig 2C, S6 Fig). Once the detected intensities were normalized to the levels of CRP, the amount of acetylated CRP in a Δ*relA* strain is 2-fold lower than WT (Fig 2C). This seems to suggest that ppGpp stimulates the acetylation of CRP under the conditions tested. As predicted, when the amount of acetylated proteins was measured in absence of CobB, the levels of acetylated CRP increased in both WT and Δ*relA* strains (S6 Fig).

**Table 1. YfiQ is required for CRP acetylation.** Amount of CRP acetylated relative to WT in absence of acetyl-phosphate and the KAT YfiQ was measured in M9 media supplemented with glycerol. Western blot can be found in S7 Fig.

|  | Acetylated CRP[1] | Amount of CRP[2] | CRP/Ac-CRP[3] |
|---|---|---|---|
| WT | $1.00 \pm 0.07$ | $1.00 \pm 0.04$ | 100% |
| Δ*ackA-pta* | $0.75 \pm 0.08$ | $1.15 \pm 0.02$ | 153% |
| Δ*ackA-pta* Δ*relA* | $0.45 \pm 0.01$ | $0.70 \pm 0.04$ | 155% |
| Δ*yfiQ* | $0.46 \pm 0.04$ | $0.90 \pm 0.01$ | 194% |
| Δ*yfiQ* Δ*relA* | $0.52 \pm 0.02$ | $0.98 \pm 0.17$ | 188% |

[1] The amount of acetylated protein was normalized to the amount of CRP protein detected in each sample.

[2] The CRP protein detected by Western blot is normalized to the amount of RpoA detected in each sample.

[3] This ratio normalizes the amount of CRP to the number of acetylated CRP.

## YfiQ is required for acetylation of CRP promoted by ppGpp

We also measured the amount of acetylated CRP in the absence of acetyl-phosphate (Δ*ackA-pta*) or in the absence of YfiQ in cells grown in M9 minimal media supplemented with glycerol (Table 1). It can be observed that in the absence of the *ackA-pta* operon, there is a slight decrease on the amount of acetylated CRP, but in the mutant Δ*ackA-pta* Δ*relA* we observe even less acetylated CRP than in a Δ*ackA-pta* mutant, suggesting that acetyl-phosphate is not responsible for the acetylation of CRP promoted by ppGpp. Interestingly, in the absence of *ackA-pta*, the expression of CRP seems to be dependent of ppGpp. A slight decrease on the levels of CRP is observed in the mutant Δ*ackA-pta* Δ*relA* (Table 1). If we consider that acetylated CRP would not bind to the promoter region, then we can normalize the amount of CRP to the amount of acetylated CRP (ratio of CRP/Ac-CRP, Table 1). We observe that both strains (Δ*ackA-pta* and Δ*ackA-pta* Δ*relA*) show similar ratios of what we may call "active CRP", which could account for the similar *lacZ* expression observed before (Fig 2A).

When we measure the relative amount of acetylated CRP (again, normalized to CRP) in a Δ*yfiQ* background are 2-fold lower than WT (Table 1), but independent of the presence or the absence of RelA, where the number of acetylated CRP does not change comparing Δ*yfiQ* with Δ*yfiQ* Δ*relA*. We interpret these results as the effect of ppGpp over CRP acetylation seems to require the presence of YfiQ, during growth in minimal media supplemented with glycerol.

## The expression of *pta*, *ackA* and *yfiQ* depends on ppGpp

Finally, we wanted to determine the effect ppGpp on the gene expression of the *ackA-pta* operon, *yfiQ* and *cobB* by qPCR (Fig 3).

The expression of *pta* show an almost 2-fold decrease in the absence of *relA* compared to WT in minimal media with glucose (Fig 3A), suggesting that the expression of *pta* depends on the presence of ppGpp. However, in minimal media with glycerol, we observe that the expression of *pta* is repressed in both strains, and the difference between WT and Δ*relA* is rendered non-significant. Instead, the expression of *ackA* (Fig 3B) seems to not be affected by ppGpp while growing in glucose. In glycerol we observe a 2-fold decrease in the expression of *ackA* in absence of *relA* compared to WT; however, as seen for *pta*, the expression of *ackA* is strongly repressed while growing in glycerol compared to glucose.

While the enzymes responsible for producing acetyl-phosphate are repressed in minimal media supplemented with glycerol (Fig 3A and 3B), the expression of *yfiQ* (Fig 3C) shows a 4-fold stimulation in cells growing in glycerol compared to glucose, consistent with CRP stimulating the expression of *yfiQ* [50]. Moreover, in absence of *relA* there is a decrease on the expression of *yfiQ* in minimal media supplemented with either glucose or glycerol (Fig 3C).

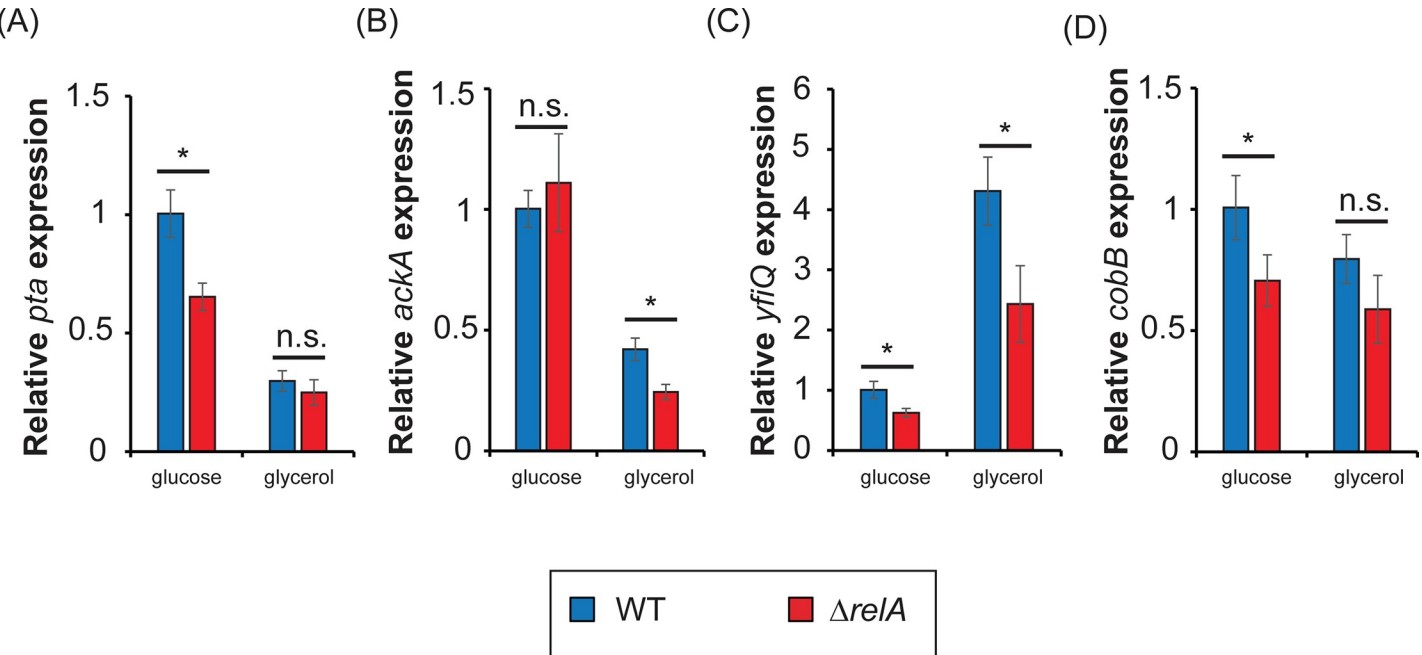

**Fig 3. Effect of ppGpp on gene expression of the main factors contributing to protein acetylation.** The gene expression of (A) *pta*, (B) *ackA*, (C) *yfiQ* and (D) *cobB* were measured by qPCR from WT and Δ*relA* cells grown in minimal media with glucose or glycerol. Expression levels are normalized to the expression in WT grown in glucose. Error bars show standard deviations of 2 biological samples and 3 technical replicates. Statistical significance was measured with T-student test (*p-value < 0.05, n.s. p-value > 0.05).

Our data suggests that ppGpp stimulates *yfiQ* expression, probably directly (considering that *yfiQ* seems to have an AT-rich discriminator). Consistently, it was previously described that the expression of *yfiQ* increases during stationary phase compared to exponential phase, but does not seem to require σ$^S$ [50]. Not many differences were observed on the expression of *cobB* (Fig 3D).

## Discussion

As previously shown [36], the gene expression of *lacZ* inversely correlates with growth rate (Fig 1A), showing lower expression in higher growth rates and higher expression in lower growth rates. This correlation has been described to be cAMP dependent [36]. Here we show that variations on the levels of ppGpp (by deleting *relA*) affect the slope of such correlation. Considering that the basal levels of ppGpp increases as growth rates decrease [35], it is perhaps not farfetched to suggest a combined effect of cAMP and ppGpp on the expression of *lacZ* and other catabolic enzymes. In fact, a similar inverse correlation of metabolic enzymes expression with growth rate was observed when the growth rate was decreased by artificially increasing the levels of ppGpp [38].

The effect of ppGpp on *lacZ* expression is not a direct effect on its promoter, but an indirect effect by controlling the acetylation of CRP. Our data suggest that during growth in minimal media supplemented with glycerol, the acetylation of CRP requires the presence of the acetyltransferases YfiQ (Fig 2B, Table 1). Both, acetyl-phosphate and YfiQ, have been shown to have the potential to acetylate certain CRP residues [51]. In that study, Kuhn et al. explore the different acetylated proteins through mass spectrometry using a WT strains and different mutants (Δ*yfiQ* among them). As they show at their supplementary material, the residue 27 of CRP seems to be acetylated in the WT strain but not in the mutant Δ*yfiQ*. Evidently, further studies

will be required to determine what residues of CRP are acetylated by YfiQ in a ppGpp-dependent manner, and how it does impair *lacZ* transcription.

As previously described, the levels of acetyl-phosphate seem to be dependent on ppGpp during growth in glucose [28], consistent with ppGpp stimulating the expression of *ackA-pta* operon in minimal media with glucose (Fig 3). Under the same conditions, acetyl-phosphate can acetylate residue K100 of CRP, modifying its ability to bind to Class II promoters, but not class I promoters [39]. There are three classes of CRP-dependent promoters, depending on the contact between CRP and the α subunit of RNAP [52]. Consequently, ppGpp has no effect on the expression *lacZ* (a class I promoter) in minimal media with glucose. In minimal media with glycerol, the expression of the operon *ackA-pta* is low (Fig 3), suggesting that the amount of acetyl-phosphate is probably low too. Instead, the levels of YfiQ dramatically increase in minimal media with glycerol, stimulated by ppGpp (Fig 3), and it can acetylate some residues of CRP, probably different from acetyl-phosphate [51]. The acetylation of CRP by YfiQ seems to be responsible for the effect of ppGpp on *lacZ* expression (Fig 2B, Table 1), suggesting that the residues acetylated by YfiQ may affect class I promoters (at least the *lac* operon). Then, we could hypothesize that while the acetylation of CRP by acetyl-phosphate during growth in glucose will affect the expression of class II promoters, the acetylation of another residue by YfiQ during growth in glycerol may affect the expression of class I promoters. Thus, by changing the acetylating state of CRP, or even the residues acetylated, ppGpp may modify the expression levels of a subset of genes regulated by CRP depending on the carbohydrate available.

Despite that deleting the *ackA-pta* operon shows a similar effect over the role of ppGpp on the *lacZ* expression than deleting *yfiQ* (Fig 2A and 2B), acetyl-phosphate does not seem to be responsible for the acetylation of CRP promoted by ppGpp in cells grown in minimal media supplemented with glycerol (Table 1). This is also consistent with the expression pattern observed (Fig 3), where the *ackA-pta* operon is repressed in media supplemented with glycerol.

As previously discussed, *in vitro* experiments using cell-free extracts–grown in media with amino acids–seem to suggest that ppGpp has an effect over the expression of the *lac* operon but only in the presence of the CRP binding site [45, 46]. Instead, similar experiments performed during amino acid starvation (by addition of pseudomonic acid) show a RelA-dependent accumulation of ppGpp that strongly inhibits the expression of *lacZ* [53]. In our case, cells are grown in minimal media without amino acids, and as observed by Little and Bremer [53], we observe that ppGpp exerts a negative effect on *lacZ* expression (increased expression in absence or RelA). However, our conditions are not as extreme as the amino acid starvation that they produce; hence, we observe a minor inhibition of *lac* operon. In any case, the effect of ppGpp on *lacZ* requires the presence of CRP [45] or the CRP binding site (S2 Fig; [46]), suggesting that it is an indirect effect by affecting CRP.

We can also find some discrepancies in the effect of ppGpp on *crp* expression: while i*n vitro* experiments suggested that ppGpp is able to repress the promoter P2 of *crp* [47], it seems that ppGpp can positively affect the expression of CRP during severe amino acid starvation [54]. Instead, no effect on the expression of CRP has been observed under the conditions tested here (S3A Fig).

Altogether, our data suggest that ppGpp can modulate the expression of genes regulated by CRP-cAMP by affecting the acetylation state of CRP, and probably, modifying its ability to bind. It is still to be determined how many genes from the ppGpp regulon depend on its effect on global protein acetylation, and more particularly, how many are shared with CRP-cAMP by changing the acetylation state of CRP. Transcriptomic studies performed during diauxic shift, show an overlap between the genes regulated by CRP and ppGpp [27], and we believe that some of these overlapping effects could be the result of CRP acetylation promoted

by ppGpp. Evidently, new transcriptomic studies that include Δ*ackA-pta* and Δ*yfiQ* mutant strains need to be performed in future studies.

## Supporting information

**S1 Fig. Levels of ppGpp.** The amount of ppGpp was measured relative to total amount of G (pppGpp + ppGpp + GTP) by TLC in MG1655 and Δ*relA* strains grown in MOPS media with 0.2% glucose or 0.4% glycerol. Error bars represent SD of two biological replicates. Statistical significance was measured with T-student test (*p-value < 0.05, n.s. p-value > 0.05).
(TIFF)

**S2 Fig. The effect of ppGpp on *lacZ* expression requires of CRP binding site.** MG1655 (WT) and CF18005 (Δ*relA*) harboring the plasmid pBbA5k were grown in M9 minimal media supplemented with glycerol 0.4% and IPTG 1 mM. Finally, fluorescence emitted by RFP (Red Fluorescent Protein) was measured. The amount of fluorescence was plotted against the OD600, giving a linear correlation where the slope is the RFP production rate per OD600. Error bars represent SD. Statistical significance was measured with T-student test (n.s. p-value > 0.05).
(TIFF)

**S3 Fig. ppGpp does not affect levels of CRP or cAMP.** (A) MG1655 (WT) and CF18005 (Δ*relA*) were grown in M9 minimal media supplemented with either glucose 0.2% or glycerol 0.4% up to exponential phase ($OD_{600}$ 0.1) and protein levels of CRP and RpoA were measured by Western blot. Means and standard deviation are shown of CRP amounts normalized to the amounts of RpoA (loading control) relative to the values from WT in glucose. (B) Measurements of cAMP of cells grown as in panel (A). Error bars show standard deviation of 2 biological samples. Statistical significance was measured with T-student test (n.s. p-value > 0.05).
(TIF)

**S4 Fig. The gene expression of $^{cyaA}$ measured by qPCR from WT and Δ*relA* in minimal media with glucose or glycerol.** Expression levels are normalized to the expression in WT grown in glucose. Error bars show standard deviation of 2 biological samples and 3 technical replicates. Statistical significance was measured with T-student test and no difference was observed between WT and Δ*relA* (p-value > 0.05).
(TIF)

**S5 Fig. The KAT YfiQ does not affect the diauxic shift.** (a) The strain MG1655 (wt) and Δ*relA*, together with their isogenic Δ*yfiQ* and Δ*cobB* mutants were grown in M9 with 0.025% glucose and 0.4% lactose for 12 h and $OD_{600}$ measured every 10 min. Ratios of diauxic time normalized to generation times of three independent experiments with duplicate wells (six values) were plotted as box plots. Bottom and top of the colored box represent first and third quartiles, and the band inside the box is the median. Whiskers represent minimum and maximum data, while circles are outliers (single points). (b) Typical diauxic growth curve from WT strain.
(TIFF)

**S6 Fig. ppGpp controls acylation of CRP, Westen blots.** The amount of acylated CRP was measured by Western blot from total extracts of WT and Δ*relA* strains together with their isogenic Δ*cobB* mutants, were grown in M9 minimal media supplemented with glycerol 0.4% up to $OD_{600}$ of 0.1. To ensure a proper identification of the CRP protein, a Δ*crp* mutant was also added. The amount of acetylated protein (detected by Western blot using antibody specific for acetylated lysines, Ac-K) was normalized to the amount of CRP and presented relative to the

WT strain. STDV = standard deviation.
(TIF)

**S7 Fig. YfiQ acetylates CRP, Western blots.** The amount of acylated CRP was measured by Western blot from total extracts of the WT strain, together with the ΔackA-pta, ΔackA-pta ΔrelA, ΔyfiQ and ΔyfiQ ΔrelA strains. Cells were grown in M9 minimal media supplemented with glycerol 0.4% up to exponential phase ($OD_{600}$ 0.1). To ensure a proper identification of the CRP protein, a Δcrp mutant was also added. The amount of acetylated protein (detected by Western blot using antibody specific for acetylated lysines, Ac-K) was normalized to the amount of CRP and presented relative to the WT strain. The CRP amounts are normalized to the amounts of RpoA (loading control).
(TIF)

**S1 Table. Bacterial strains and plasmids used in this study.**
(PDF)

**S2 Table. List of primers used in this report.**
(PDF)

**S1 File. Uncropped and unadjusted Western blot images.**
(PDF)

## Acknowledgments

We would like to thank Dr. Sankar Adhya (NCI) for providing material necessary for the construction of different strains.

## Author Contributions

**Conceptualization:** Llorenç Fernández-Coll.

**Formal analysis:** Chunghwan Ro, Llorenç Fernández-Coll.

**Funding acquisition:** Michael Cashel.

**Investigation:** Chunghwan Ro, Llorenç Fernández-Coll.

**Project administration:** Llorenç Fernández-Coll.

**Supervision:** Michael Cashel, Llorenç Fernández-Coll.

**Validation:** Chunghwan Ro, Michael Cashel, Llorenç Fernández-Coll.

**Visualization:** Chunghwan Ro, Llorenç Fernández-Coll.

**Writing – original draft:** Chunghwan Ro.

**Writing – review & editing:** Chunghwan Ro, Michael Cashel, Llorenç Fernández-Coll.

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
