## [Decision Letter · Decision Letter 0]

13 Jul 2021

PONE-D-21-19172

The secondary messenger ppGpp interferes with cAMP-CRP regulon by promoting CRP acetylation in Escherichia coli.

PLOS ONE

Dear Dr. Fernández-Coll,

Thank you for submitting your manuscript to PLOS ONE. After careful consideration, we feel that it has merit but does not fully meet PLOS ONE’s publication criteria as it currently stands. Therefore, we invite you to submit a revised version of the manuscript that addresses the points raised during the review process.

The reviewers agree that this is a well written and interesting manuscript; however, one of the reviewers is concerned that the data presented does not support the conclusions drawn here. Because this will require another round of review, I have designated this manuscript as a major revision. Both reviewers have provided specific points that they believe will improve the manuscript, please address each of these points in your resubmission. 

We look forward to receiving your revised manuscript.

Kind regards,

Benjamin J. Koestler, Ph.D.

Academic Editor

PLOS ONE

Journal Requirements:

Reviewers' comments:

Reviewer's Responses to Questions

**Comments to the Author**

1. Is the manuscript technically sound, and do the data support the conclusions?

Reviewer #1: Yes

Reviewer #2: Partly

2. Has the statistical analysis been performed appropriately and rigorously? 

Reviewer #1: Yes

Reviewer #2: No

3. Have the authors made all data underlying the findings in their manuscript fully available?

Reviewer #1: Yes

Reviewer #2: Yes

4. Is the manuscript presented in an intelligible fashion and written in standard English?

Reviewer #1: Yes

Reviewer #2: Yes

5. Review Comments to the Author

Reviewer #1: Nucleotide second messengers (NSMs) are utilized across all domains of life. Bacteria respond to environmental stimuli by increasing second messengers, each of which transmits and amplifies the signal by activating a subset of downstream pathways to optimize appropriate physiological adaptations. Each NSM has multiple but distinct downstream targets. In this nicely written manuscript, Ro et al. address an indirect crosstalk between two second messenger systems. They show that ppGpp interferes with expression of the cAMP-CRP regulon by stimulating expression of the acetyltransferase YfiQ which acetylates CRP in Escherichia coli. This interesting study may contribute to the base of our knowledge on bacterial physiology. I have only one suggestion to offer concerning the physiological role of this regulation.

The data in this study suggest that ppGpp stimulates expression of YfiQ and thereby increases CRP acetylation (Figures 2, 3, & S5). The acetylation of CRP impairs its ability to bind to DNA. Therefore, according to this regulation, ppGpp interferes with lacZ expression. However, as previously known that both the expression of the CRP-cAMP regulon and the levels of ppGpp inversely correlates with the growth rate, authors observed higher levels of beta-galactosidase activities and ppGpp in glycerol than in glucose (Figures 1 and S1). Thus, the higher level of ppGpp appears to correlate with the higher expression of the CRP regulon gene, lacZ, under certain circumstances. Therefore, the physiological significance of the CRP-interfering role of ppGpp needs to be more clearly stated in the discussion session. In what circumstances would this regulation operate?

Minor corrections:

1. L95. N-acetylglucosamine (A in lower case).

2. L97, 1 mM (space).

3. L134. Is there any specific reason why he scpA gene was used as an internal control, instead of a house-keeping gene such a rRNA-coding gene?

4. L137. INTEReFERES.

5. L164. The maximum beta-galactosidase activity?

6. L174. In Fig S2B, RFP, but not lacZ, was used as a reporter gene.

7. L177. acETylation?

8. L261. showS.

9. L289. We could not find that YfiQ has the potential to acetylate certain CRP residues in the reference 50.

Reviewer #2: Manuscript (PONE-D-21-19172): The secondary messenger ppGpp interferes with cAMP-CRP regulon by promoting CRP acetylation in Escherichia coli.

Authors: Ch. Ro, M. Cashel and Ll. Fernández-Coll

In this manuscript Ro et al., investigated the relationship between the alarmone ppGpp, the Catabolite Represor Protein CRP and the second messenger cAMP in regulating the expresión of the lacZ gene from Escherichia coli. Employing genetic, physiologic and immunologic approaches, the authors described that i) ppGpp indirectly affects the expression of lacZ by affecting the CRP-cAMP complex, ii) ppGpp regulates the expression lacZ through CRP-acetylation, iii) YfiQ is responsible of acetylating CRP in a RelA-dependent manner, and finally, iv) RelA regulates in a positive manner the expression of yfiQ. Based on these results, they concluded that ppGpp induce the acetylation of CRP, thus affecting its regulatory properties with a direct impact on the expression of the lacZ gene.

The work addresses a very interesting topic regarding a novel posttranslational mechanism that presumably regulates the transcriptional properties of cAMP-CRP over the Lac operon in E. coli. However, the experimental evidence shown in only 3 Figures in the main text is not enough to support this notion. Below, please find comments and suggestions that the authors may find useful to improve the scientific impact of their manuscript.

Major comments:

1) Lines 138-139: In the experimental approach the authors determined beta-galactosidase levels to evaluate the transcriptional control exerted by cAMP-CRP over the lacZ operon. In these paragraphs they indicate that addition of IPTG (1 mM) eliminate the repressor effects of LacI; however, in most of the experiments the authors don’t indicated if the minimal media employed with different carbon sources was supplemented with IPTG. Furthermore, with such amenable model of study, it calls the attention that the authors did not employ a lacI-deficient strain to unambiguously rule out the repressor effects of LacI.

2) Lines 146-51: The authors justify the advantage of using a RelA-deficient strain instead of a double relA spoT mutant. The rational is correct; however, as shown in Figure S1, in culture media supplemented with glycerol and glucose the relA mutant still contain ppGpp. Can the authors clarify if this amount of the alarmone is influencing or not their results. Perhaps, the use of an available relA spoT strain would render more confident experiments in a genetic background fully depleted of ppGpp. Also, Figure S1 do not show if there are statistical differences in the levels of ppGpp between the strains and carbon sources employed, please include this information in the revised manuscript.

3) As the authors stated in the introduction, the cAMP-CRP complex controls the expression of several catabolic genes; therefore, to better reinforce the conclusions described in this work, the authors must include experiments with reporters additional to lacZ.

4) Lines 165-168. The manner of presenting the results in Figure 1 are rather confuse. This reviewer and the interested audience would be benefited if the authors present in a more detailed form these results. Please move the results of Figure S2-A to the main text with the following considerations:

A) It is not clear the growth phase in which the strains were analyzed; therefore, these plots must show the growth levels by OD600nm and ideally by viable counts.

B) In the same Figure, plot in a time dependent fashion, the levels of beta-galactosidase of the strains analyzed in the distinct carbon sources employed

C) Here, a question arises: what is the functional relationship between RelA and Crp in regulating LacZ levels in a LacI-deficient background? This question can be easily tested in relA crp genetic background but plotting the results in a time dependent manner.

5. Lines 169-176:

5.1. Lines 169-170. Please be more specific in explaining how the authors expect that ppGpp could affect the lacZ promoter; more specifically, is not expected that ppGpp regulates lacZ (RFP in their experiments) expression by binding RNA polymerase? Please see https://doi.org/10.1073/pnas.1819682116

5.2. In the results of Figure S2-B, please make sure to explain:

A) that you are using a PlacUV5-Rfp fusion in an extrachromosomal multi-copy plasmid

B) the experiment does not directly measure lacZ, but RFP expression.

C) with a single experimental time-point is difficult to verify that ppGpp (RelA deficiency) does not affect lac-RFP expression, these experiments must be performed in time-dependent fashion plotting also the growth curve, ideally as indicated in the next comment

D) to avoid effects of gene dosage and more appropriately address these experiments, the authors must have considered to work with a strain carrying a single chromosomal copy of the PlacUV5-lacZ fusion.

6. Lines 178-184. Here, the authors stated that the absence of RelA has no effect on the levels of cAMP-CRP. To support this notion, they, independently, analyzed protein levels of CRP, cAMP concentrations and mRNA levels of cyaA. However, to better support this conclusion the authors are strongly encouraged to determine the levels of the cAMP-CRP complex.

7. CRP acetylation experiments (Fig. 2; Figs. S5, S6 and Table 1)

A) Figure 2. Notation of strains tested in Fig. 2 is confuse, please clearly indicate the genotype of each strain and present the results in the following order: WT, ackA-pta, relA, relA/ackA-pta establishing significant differences among the four strains. Please, describe and interprete these comparisons in the main text of the paper.

B). Figs S5 and S6 and Table 1. With the immunological (Western blotting) approach employed to determine levels of acetylated CRP in the distinct genetic backgrounds, is a little difficult to agree with the conclusion of the authors that relA, ackA-pta, yfiQ and cobB regulate the levels of acetylated CRP. Even in the absence of RelA and YfiQ or RelA and AckA-Pta, the anti Ac-K antobodie detects bands of similar intensity than those observed in the WT strain. Furthermore, the protein band control (which I assume is RpoA) shows different intensities. Importantly, a significant fraction of acetylated CRP is detected in all the genetic strains analyzed making more difficult to interprete these results. Therefore, the authors are strongly encouraged to find alternative strategies to quantitate levels of acetylated CRP for instance by LC–MS/MS analysis.

8. The conclusion that YfiQ is responsible of acetylating CRP must be clearly documented as this is a key evidence of the regulation mechanism of CRP proposed by the authors. Please specifiy if CRP has been described as a target for YfIQ. If not, the authors are encouraged to implement a more specific approach to demonstrate this fact, for example, using in in vitro assays the unacetylated form of pure CRP as a substrate for the purified form of YfiQ.

9. Finally, regarding the conclusion that yfiQ expression is regulated by the levels of ppGpp (Fig. 3), the authors must indicate in the discussion section how the alarmone may exert its regulatory functions.

Minor comments.

1) Lines 98-101: Please describe in more detail this protocol, indicate the substrate used and specify how the Miller Units were determined.

2) Lines 123-126. Please show as a supplementary Figure, a typical growth curve to see the diauxic effect and clearly indicate in this curve the OD600nm at which most of the sample were taken for biochemical and immunological determinations.

3) Figs. S3 and S6. Please describe in the Figure Legends the purpose of the Western Blotting of RpoA in these experiments.

6. PLOS authors have the option to publish the peer review history of their article (what does this mean?). If published, this will include your full peer review and any attached files.

Reviewer #1: **Yes: **Yeong-Jae Seok

Reviewer #2: No

---

## [Author Response · Author response to Decision Letter 0]

20 Sep 2021

We have upload a response to the reviewers containing the comments of the reviewers (in black) and our response (in blue). Here we submit only our response to the reviewers' commets. We also want to thank the reviewers for their comments. 

Reviewer #1: 

We would like to say that this is a really interesting question and this discussion has been added to the modified version of the manuscript (lines 319-335). As previously said in the manuscript, the levels of acetyl-phosphate seem to be dependent of ppGpp during growth in glucose (Fernandez-coll and Cashel 2018), consistent with ppGpp stimulating the expression of ackA-pta operon in minimal media with glucose (figure 3). Under the same conditions, acetyl-phosphate can acetylate CRP, modifying its ability to bind to Class 2 promoters, but not class 1 promoters (Davis et al. 2018). Consequently, ppGpp has no effect on the expression lacZ (a class 1 promoter) in minimal media with glucose. In minimal media with glycerol, the expression of the operon ackA-pta is low (figure 3), suggesting that the amount of acetyl-phosphate is probably low too. Instead, the levels of YfiQ dramatically increase in minimal media with glycerol, stimulated by ppGpp (figure 3), and it can acetylate some residues of CRP, different from acetyl-phosphate (Kuhn et al. 2014). The acetylation of CRP by YfiQ seems to be responsible for the effect of ppGpp on lacZ expression (figure 2B, Table 1), suggesting that the residues acetylated by YfiQ may affect class 1 promoters (at least the lac operon). Then, we could hypothesize that while the acetylation of CRP by acetyl-phosphate during growth in glucose will affect the expression of class 2 promoters, the acetylation of another set of residues by YfiQ during growth in glycerol may affect the expression of class 1 promoters. It is possible that by changing the acetylating state of CRP, or even the residues acetylated, ppGpp may modify the expression levels of a subset of genes regulated by CRP depending on the carbohydrate available.

Minor corrections:

1. and 2. Changed as suggested.

3. Considering that we were planning to study the effects of ppGpp and CRP over gene regulation, we needed a gene that was not affected by either. According to Traxler et al. 2006, scpA is not affected by ppGpp or CRP. Instead, the rRNA operons and other commonly used house-keeping genes (gyrA, parC, zwf, gapA), are regulated by ppGpp or CRP. This is now explained in the modified version of the manuscript (lines 139-140).

4 and 5 changed as suggested. 

6. This is now indicated in the text (line 191). 

7. and 8. Changed as suggested

9. At Kuhn et al, the authors determine the amount of acetylated proteins through mass spectrometry of WT strains and different mutants (yfiQ among them). As they show at table S1, residue 26 seems to be acetylated in the WT strain, but not in the mutant yfiQ. This information is now described in the discussion (lines 314-318). 

Reviewer #2: 

Major comments:

1) As indicated by You et al. 2013, adding 1 mM of IPTG mimics a LacI-deficient background, having no effect on the correlation between lacZ levels and growth rate. Thus we decided to not include it on our manuscript, because we believe that it has been already proven. This is now stressed in the text (lines 147-148).

2) As mentioned in line 157, a mutant ΔrelA ΔspoT will not grow in the conditions used for this study, minimal media without amino acids (Xiao, et al. 1991). That forces us, as well as other researchers in our field, to use the closest strain possible to a ppGpp0 strain: in this case a ΔrelA strain. RelA-dependent synthesis of ppGpp responds to starvation of amino acids, but also it will be stimulated by starvation of precursors of amino acids (responding to different carbon sources). Therefore, under the tested conditions, most of the ppGpp is synthesized by RelA, and consequently, a strain ΔrelA shows almost undetectable levels of ppGpp (fig. S1). This is further discussed in the revised manuscript (lines 157-164). Moreover, the statistical differences have been included at the figure S1, as requested by the reviewer. 

3) We agree with the reviewer that more reporter genes would strengthen our point, the more the merrier. However, we also need to stress that lacZ has been a staple to report effects over the CRP complex for decades. Transcriptomic studies have shown an overlap between the genes regulated by CRP and ppGpp (Traxler et al. 2006), and we believe that our results suggest that some of these overlapping effects could be the result of CRP acetylation promoted by ppGpp. Therefore, to address this hypothesis, new transcriptomic studies that include ΔackA-pta and ΔyfiQ mutant strains are required. Evidently, this will be part of a follow up manuscript. This is now discussed in the manuscript (line 359-362). 

4):

A) and B) The figure 1 is based on the experiments described by You et al. 2013, as indicated on the text, thus we decided to plot it in the same way. There the authors show a correlation between growth rate (μ) and lacZ expression. Then, we believe that plotting OD600nm and β-galactosidase activity in a time-dependent manner may confuse the reader from the main point of the figure, that the β-galactosidase activity is directly proportional to the growth rate and this can be modified by ppGpp. We agree that Figure S2A will help on the comprehension of the figure, and we have moved it to the main text, as suggested by the reviewer. All the samples for measuring β-galactosidase activity were taken at OD600nm of 0.1. Now, this is also added to the figure legend. The growth levels for each strain are indicated with the growth rates (μ) calculated with the OD600nm. 

C) This is an interesting point raised by the reviewer. As previously discussed, You et al. 2013 show that the addition of IPTG mimics a LacI-background, and does not affect the regulation of LacZ levels. So, we consider that addition of 1 mM of IPTG has a similar effect as a LacI-deficient background. As mentioned above, this is now stressed in the manuscript. We also agree with the reviewer that a classic way to determine the involvement of CRP on the regulation of ppGpp on lacZ expression, it would be the use of a Δcrp mutant strain. However, a Δcrp strain does not express lacZ (undetectable β-galactosidase activity) and it does not grow in minimal media supplemented with glycerol. Therefore, we decided to include the experiments using the lacUV5 promoter (that do not contain the CRP-binding site), despite it is just a confirmation of a previous result (Yang et al. 1974). 

5. Lines 169-176:

5.1. It has previously shown (Yang et al. 1974) that lacUV5 does not respond to the levels of ppGpp. Actually, it has been used by several authors as negative control in experiments of in vitro transcription when studying ppGpp regulation. Then, the experiments in figure S2 need to be taken as a verification of a previous result. We have stressed this notion in the reviewed version of the manuscript (line 191). As pointed by the reviewer, genes directly regulated by ppGpp show a GC-rich discriminator (region between -10 box and +1) if repressed by ppGpp or an AT-rich discriminator if stimulated. Neither lacUV5 nor lac promoter show such discriminator. This information is now added to the manuscript (lines 185-188). 

5.2. 

A) and B) This has been mentioned in the modified version of the manuscript (line 191), and the figure legend.

C) We agree with the reviewer that this kind of experiments require more than one time-point, and we apologize for the lack of clarity in the figure. In fact, in figure S2 we are not representing the fluorescence of a single time-point, but the rate of production per OD600nm (slope obtained after plotting fluorescence accumulation vs. OD600nm). This is now mentioned in the modified version. However, considering that this experiment is the verification of something previously described (Yang et al. 1974), we do not think that it is necessary to add extra plots or the growth curve.

D) As previously mentioned, this was a confirmation of a published result (Yang et al. 1974). Thus, we believe that it was well established in the original paper. 

6. We agree with the reviewer that we determine the effect of ppGpp over the levels of CRP and cAMP but not the levels of the cAMP-CRP complex. If we mentioned on the text that “ppGpp do not affect the cAMP-CRP complex” it has been for a matter of simplicity. This has been modified in the text accordingly. About the recommendation to measure the levels of the complex, first, we do not have the capacity or the experience to perform such experiment. And second, there is no evidence to suggest that ppGpp can physically interfere with the binding of cAMP to CRP. Despite that ppGpp can bind to some proteins, most of the effects are through changing gene expression. Few protein interactions with ppGpp have been described lately, but none with CRP. There are no evidences either that ppGpp can directly interact with cAMP. It is arguable that the acetylation of CRP could interfere with the formation of the complex with cAMP, but as discussed in the text (lines 317-318), that will have to be determined in future studies. 

7. A) The aim of the figure 2 is to show that the effect of ppGpp over lacZ requires the acetylation of CRP. First, we have to clarify that panel A and B show β-galactosidase activity experiments, while panel C show CRP’s acetylation levels only in the presence or the absence of RelA. This is now stressed in the figure legend. 

Second, the nomenclature used in this figure is the same as used in the rest of the manuscript to ensure the cohesion of the manuscript. In panel A and B, the presence of RelA is indicated by color (as indicated at the figure legend). Any other genetic background is indicated in the X-axis. In panel C, it is true that the color code is broken (both columns are black), but there is only one category. 

Third, the acetylation of CRP in ΔackA-pta and ΔyfiQ mutant strains is shown only in Table1 (discussed below). 

B) Figures S6 and S7 contain the Western blots to determine the levels of acetylated CRP shown in Figure 2C and Table 1, respectively. We agree that the anti Ac-K antibody detects bands that, at naked eye, seem to be of similar intensity but, those intensities were normalized to the amount of CRP detected at the same samples. Moreover, the intensities are not measured de visu but through the software Image Studio from Li-Cor. This information is now added to the manuscript (line 113-114). Once the detected intensities were normalized to the levels of CRP, the amount of acetylated CRP in a ΔrelA strain happen to be 2-fold lower than WT, and this is reported in fig. 2C. As we interpret this result, ppGpp seems to stimulate the acetylation of CRP under the conditions tested. In figure S7, showing the Western blots used for the table 1, we are determining the effects of YfiQ and AckA-Pta over the levels of CRP (normalized to RpoA) and the levels of acetylated CRP (normalized to the levels of CRP). Here, we observe that the measured intensities of acetylated CRP (again, normalized to CRP) in a ΔyfiQ background are 2-fold lower than WT, but independent of the presence or the absence of RelA. We interpret these results as the effect of ppGpp over CRP acetylation seems to require the presence of YfiQ, and this is how it is now stated on the text. 

Despite that LC-MS/MS will give us quantitative levels of acetylated CRP, the antibody used in this manuscript have been used to obtain relative amounts of acetylated proteins in manuscript previously published by our research group, and other research groups in our field. We have stressed that our results are relative amounts instead of quantitative levels (line 247). 

8. Acetylation of CRP, yfiQ-dependent, have been previously documented by Kuhn et al. 2014. This has been further discussed, as requested by reviewer 1 (as can be seen above, lines 314-317). It is true that further studies are required to determine more in detail what residues are acetylatilated in a ppGpp-dependent manner, and how does it impair lacZ transcription. But, as it is now discussed in the modified text (lines 317-318), this will be the focus of future manuscripts. 

9. We absolutely agree with the reviewer. As previously said, genes directly stimulated by ppGpp has an A/T rich discriminator. As shown by Castaño-Cerezo et al. 2011, the expression of yfiQ is stimulated in stationary phase, but it does not respond to the alternative sigma factor RpoS, suggesting that any effect must be direct. This is now discussed at the modified version of the manuscript (lines 297-299).

Minor comments.

1) This protocol is a well established standard in our field, but, as suggested by the reviewer, more information has been added. 

2) We agree that adding a diauxic growth curve will help to understand some parts of the manuscript; it is now a panel of figure S5. However, we believe that indicating there the OD at which most of the samples were taken will be misleading for the reader, considering that only the experiment shown in figure S5 was made during diauxic shift. Instead, we have stressed in all the figure legends that the samples were taken during exponential phase (at an OD600nm of 0.1). 

3) This is now added to the figure legends as suggested by the reviewer.

---

## [Decision Letter · Decision Letter 1]

12 Oct 2021

The secondary messenger ppGpp interferes with cAMP-CRP regulon by promoting CRP acetylation in Escherichia coli.

PONE-D-21-19172R1

Dear Dr. Fernández-Coll,

We’re pleased to inform you that your manuscript has been judged scientifically suitable for publication and will be formally accepted for publication once it meets all outstanding technical requirements.

Kind regards,

Benjamin J. Koestler, Ph.D.

Academic Editor

PLOS ONE

Additional Editor Comments (optional):

Reviewers' comments:

Reviewer's Responses to Questions

**Comments to the Author**

1. If the authors have adequately addressed your comments raised in a previous round of review and you feel that this manuscript is now acceptable for publication, you may indicate that here to bypass the “Comments to the Author” section, enter your conflict of interest statement in the “Confidential to Editor” section, and submit your "Accept" recommendation.

Reviewer #1: All comments have been addressed

Reviewer #2: All comments have been addressed

2. Is the manuscript technically sound, and do the data support the conclusions?

Reviewer #1: Yes

Reviewer #2: Yes

3. Has the statistical analysis been performed appropriately and rigorously? 

Reviewer #1: Yes

Reviewer #2: Yes

4. Have the authors made all data underlying the findings in their manuscript fully available?

Reviewer #1: Yes

Reviewer #2: Yes

5. Is the manuscript presented in an intelligible fashion and written in standard English?

Reviewer #1: Yes

Reviewer #2: Yes

6. Review Comments to the Author

Reviewer #1: (No Response)

Reviewer #2: Manuscript PONE-D-21-19172R1.

The secondary messenger ppGpp interferes with cAMP-CRP regulon by promoting CRP acetylation in Escherichia coli.

Authors: Ch. Ro, M. Cashel and Ll. Fernández-Coll

In this manuscript Ro et al., investigated the relationship between the alarmone ppGpp, the Catabolite Represor Protein CRP and the second messenger cAMP in regulating the expresión of the lacZ gene from Escherichia coli. Employing genetic, physiologic and immunologic approaches, the authors described that i) ppGpp indirectly affects the expression of lacZ by affecting the CRP-cAMP complex, ii) ppGpp regulates the expression lacZ through CRPacetylation, iii) YfiQ is responsible of acetylating CRP in a RelA-dependent manner, and finally, iv) RelA regulates in a positive manner the expression of yfiQ. Based on these results, they concluded that ppGpp induce the acetylation of CRP, thus affecting its regulatory properties

with a direct impact on the expression of the lacZ gene.

Below please find my comments to the author´s responses.

Major comments:

1) Lines 138-139: In the experimental approach the authors determined beta-galactosidase levels to evaluate the transcriptional control exerted by cAMP-CRP over the lacZ operon. In these paragraphs they indicate that addition of IPTG (1 mM) eliminate the repressor effects of LacI; however, in most of the experiments the authors don’t indicate if the minimal media employed with different carbon sources was supplemented with IPTG. Furthermore, with such amenable model of study, it calls the attention that the authors did not employ a lacI-deficient strain to unambiguously rule out the repressor effects of LacI.

As indicated by You et al. 2013, adding 1 mM of IPTG mimics a LacI-deficient background, having no effect on the correlation between lacZ levels and growth rate. Thus we decided to not include it on our manuscript, because we believe that it has been already proven. This is now stressed in the text (lines 147-148).

Reviewer comment (RC): Comment appropriately addressed.

2) Lines 146-51: The authors justify the advantage of using a RelA-deficient strain instead of a double relA spoT mutant. The rational is correct; however, as shown in Figure S1, in culture media supplemented with glycerol and glucose the relA mutant still contain ppGpp. Can the authors clarify if this amount of the alarmone is influencing or not their results. Perhaps, the use of an available relA spoT strain would render more confident experiments in a genetic background fully depleted of ppGpp. Also, Figure S1 do not show if there are statistical differences in the levels of ppGpp between the strains and carbon sources employed, please include this information in the revised manuscript.

As mentioned in line 157, a mutant ΔrelA ΔspoT will not grow in the conditions used for this study, minimal media without amino acids (Xiao, et al. 1991). That forces us, as well as other researchers in our field, to use the closest strain possible to a ppGpp0 strain: in this case a ΔrelA strain. RelA-dependent synthesis of ppGpp responds to starvation of amino acids, but also it will be stimulated by starvation of precursors of amino acids (responding to different carbon sources). Therefore, under the tested conditions, most of the ppGpp is synthesized by RelA, and consequently, a strain ΔrelA shows almost undetectable levels of ppGpp (fig. S1). This is further discussed in the revised manuscript (lines 157-164). Moreover, the statistical differences have been included at the figure S1, as requested by the reviewer.

RC: Comment appropriately addressed.

3) As the authors stated in the introduction, the cAMP-CRP complex controls the expression of several catabolic genes; therefore, to better reinforce the conclusions described in this work, the authors must include experiments with reporters additional to lacZ.

We agree with the reviewer that more reporter genes would strengthen our point, the more the merrier. However, we also need to stress that lacZ has been a staple to report effects over the CRP complex for decades. Transcriptomic studies have shown an overlap between the genes regulated by CRP and ppGpp (Traxler et al. 2006), and we believe that our results suggest that some of these overlapping effects could be the result of CRP acetylation promoted by ppGpp. Therefore, to address this hypothesis, new transcriptomic studies that include ΔackA-pta and ΔyfiQ mutant strains are required. Evidently, this will be part of a follow up manuscript. This is now discussed in the manuscript (line 359-362).

RC: Comment appropriately addressed.

4) Lines 165-168. The manner of presenting the results in Figure 1 are rather confuse. This reviewer and the interested audience would be benefited if the authors present in a more detailed form these results. Please move the results of Figure S2-A to the main text with the following considerations:

A) It is not clear the growth phase in which the strains were analyzed; therefore, these plots must show the growth levels by OD600nm and ideally by viable counts.

B) In the same Figure, plot in a time dependent fashion, the levels of beta-galactosidase of the strains analyzed in the distinct carbon sources employed

The figure 1 is based on the experiments described by You et al. 2013, as indicated on the text, thus we decided to plot it in the same way. There the authors show a correlation between growth rate (μ) and lacZ expression. Then, we believe that plotting OD600nm and β-galactosidase activity in a time-dependent manner may confuse the reader from the main point of the figure, that the β-galactosidase activity is directly proportional to the growth rate and this can be modified by ppGpp. We agree that Figure S2A will help on the comprehension of the figure, and we have moved it to the main text, as suggested by the reviewer. All the samples for measuring β-galactosidase activity were taken at OD600nm of 0.1. Now, this is also added to the figure legend. The growth levels for each strain are indicated with the growth rates (μ) calculated with the OD600nm.

RC: Comment addressed.

C) Here, a question arises: what is the functional relationship between RelA and Crp in regulating LacZ levels in a LacI-deficient background? This question can be easily tested in relA crp genetic background but plotting the results in a time dependent manner.

This is an interesting point raised by the reviewer. As previously discussed, You et al. 2013 show that the addition of IPTG mimics a LacI-background, and does not affect the regulation of LacZ levels. So, we consider that addition of 1 mM of IPTG has a similar effect as a LacIdeficient background. As mentioned above, this is now stressed in the manuscript. We also agree with the reviewer that a classic way to determine the involvement of CRP on the regulation of ppGpp on lacZ expression, it would be the use of a Δcrp mutant strain. However, a Δcrp strain does not express lacZ (undetectable β-galactosidase activity) and it does not grow in minimal media supplemented with glycerol. Therefore, we decided to include the experiments using the lacUV5 promoter (that do not contain the CRP-binding site), despite it is just a confirmation of a previous result (Yang et al. 1974).

RC: Ok

5. Lines 169-176:

5.1. Lines 169-170. Please be more specific in explaining how the authors expect that ppGpp could affect the lacZ promoter; more specifically, is not expected that ppGpp regulates lacZ (RFP in their experiments) expression by binding RNA polymerase? Please see https://doi.org/10.1073/pnas.1819682116

It has previously shown (Yang et al. 1974) that lacUV5 does not respond to the levels of ppGpp. Actually, it has been used by several authors as negative control in experiments of in vitro transcription when studying ppGpp regulation. Then, the experiments in figure S2 need to be taken as a verification of a previous result. We have stressed this notion in the reviewed version of the manuscript (line 191). As pointed by the reviewer, genes directly regulated by ppGpp show a GC-rich discriminator (region between -10 box and +1) if repressed by ppGpp or an Atrich discriminator if stimulated. Neither lacUV5 nor lac promoter show such discriminator. This information is now added to the manuscript (lines 185-188).

RC: Ok

5.2. In the results of Figure S2-B, please make sure to explain:

A) that you are using a PlacUV5-Rfp fusion in an extrachromosomal multi-copy plasmid

B) the experiment does not directly measure lacZ, but RFP expression.

This has been mentioned in the modified version of the manuscript (line 191), and the figure legend.

RC: Ok

C) with a single experimental time-point is difficult to verify that ppGpp (RelA deficiency) does not affect lac-RFP expression, these experiments must be performed in time-dependent fashion plotting also the growth curve, ideally as indicated in the next comment.

We agree with the reviewer that this kind of experiments require more than one time-point, and we apologize for the lack of clarity in the figure. In fact, in figure S2 we are not representing the fluorescence of a single time-point, but the rate of production per OD600nm (slope obtained after plotting fluorescence accumulation vs. OD600nm). This is now mentioned in the modified version. However, considering that this experiment is the verification of something previously described (Yang et al. 1974), we do not think that it is necessary to add extra plots or the growth curve.

RC: Ok

D) to avoid effects of gene dosage and more appropriately address these experiments, the authors must have considered to work with a strain carrying a single chromosomal copy of the PlacUV5-lacZ fusion.

As previously mentioned, this was a confirmation of a published result (Yang et al. 1974). Thus, we believe that it was well established in the original paper.

RC: Ok

6. Lines 178-184. Here, the authors stated that the absence of RelA has no effect on the levels of cAMP-CRP. To support this notion, they, independently, analyzed protein levels of CRP, cAMP concentrations and mRNA levels of cyaA. However, to better support this conclusion the authors are strongly encouraged to determine the levels of the cAMP-CRP complex.

We agree with the reviewer that we determine the effect of ppGpp over the levels of CRP and cAMP but not the levels of the cAMP-CRP complex. If we mentioned on the text that “ppGpp do not affect the cAMP-CRP complex” it has been for a matter of simplicity. This has been modified in the text accordingly. About the recommendation to measure the levels of the complex, first, we do not have the capacity or the experience to perform such experiment. And second, there is no evidence to suggest that ppGpp can physically interfere with the binding of cAMP to CRP. Despite that ppGpp can bind to some proteins, most of the effects are through changing gene expression. Few protein interactions with ppGpp have been described lately, but none with CRP. There are no evidences either that ppGpp can directly interact with cAMP. It is arguable that the acetylation of CRP could interfere with the formation of the complex with cAMP, but as discussed in the text (lines 317-318), that will have to be determined in future studies.

RC: Ok

7. CRP acetylation experiments (Fig. 2; Figs. S5, S6 and Table 1)

A) Figure 2. Notation of strains tested in Fig. 2 is confuse, please clearly indicate the genotype of each strain and present the results in the following order: WT, ackA-pta, relA, relA/ackA-pta establishing significant differences among the four strains. Please, describe and interprete these comparisons in the main text of the paper.

The aim of the figure 2 is to show that the effect of ppGpp over lacZ requires the acetylation of CRP. First, we have to clarify that panel A and B show β-galactosidase activity experiments, while panel C show CRP’s acetylation levels only in the presence or the absence of RelA. This is now stressed in the figure legend.

Second, the nomenclature used in this figure is the same as used in the rest of the manuscript to ensure the cohesion of the manuscript. In panel A and B, the presence of RelA is indicated by color (as indicated at the figure legend). Any other genetic background is indicated in the Xaxis. In panel C, it is true that the color code is broken (both columns are black), but there is only one category. Third, the acetylation of CRP in ΔackA-pta and ΔyfiQ mutant strains is shown only in Table1 (discussed below).

RC: Ok

B). Figs S5 and S6 and Table 1. With the immunological (Western blotting) approach employed to determine levels of acetylated CRP in the distinct genetic backgrounds, is a little difficult to agree with the conclusion of the authors that relA, ackA-pta, yfiQ and cobB regulate the levels of acetylated CRP. Even in the absence of RelA and YfiQ or RelA and AckA-Pta, the anti Ac-K antobodie detects bands of similar intensity than those observed in the WT strain. Furthermore, the protein band control (which I assume is RpoA) shows different intensities. Importantly, a significant fraction of acetylated CRP is detected in all the genetic strains analyzed making more difficult to interprete these results. Therefore, the authors are strongly encouraged to find alternative strategies to quantitate levels of acetylated CRP for instance by LC–MS/MS analysis.

Figures S6 and S7 contain the Western blots to determine the levels of acetylated CRP shown in Figure 2C and Table 1, respectively. We agree that the anti Ac-K antibody detects bands that, at naked eye, seem to be of similar intensity but, those intensities were normalized to the amount of CRP detected at the same samples. Moreover, the intensities are not measured de visu but through the software Image Studio from Li-Cor. This information is now added to the manuscript (line 113-114). Once the detected intensities were normalized to the levels of CRP, the amount of acetylated CRP in a ΔrelA strain happen to be 2-fold lower than WT, and this is reported in fig. 2C. As we interpret this result, ppGpp seems to stimulate the acetylation of CRP under the conditions tested. In figure S7, showing the Western blots used for the table 1, we are determining the effects of YfiQ and AckA-Pta over the levels of CRP (normalized to RpoA) and the levels of acetylated CRP (normalized to the levels of CRP). Here, we observe that the measured intensities of acetylated CRP (again, normalized to CRP) in a ΔyfiQ background are 2-fold lower than WT, but independent of the presence or the absence of RelA. We interpret these results as the effect of ppGpp over CRP acetylation seems to require the presence of YfiQ, and this is how it is now stated on the text.

Despite that LC-MS/MS will give us quantitative levels of acetylated CRP, the antibody used in this manuscript have been used to obtain relative amounts of acetylated proteins in manuscript previously published by our research group, and other research groups in our field. We have stressed that our results are relative amounts instead of quantitative levels (line 247).

RC: Experiments shown in Figs S6 and S7 are relevamt to support a major conclusion of the manuscript, therefore they must be shown in the main text.

8. The conclusion that YfiQ is responsible of acetylating CRP must be clearly documented as this is a key evidence of the regulation mechanism of CRP proposed by the authors. Please specifiy if CRP has been described as a target for YfIQ. If not, the authors are encouraged to implement a more specific approach to demonstrate this fact, for example, using in in vitro assays the unacetylated form of pure CRP as a substrate for the purified form of YfiQ.

Acetylation of CRP, yfiQ-dependent, have been previously documented by Kuhn et al. 2014. This has been further discussed, as requested by reviewer 1 (as can be seen above, lines 314- 317). It is true that further studies are required to determine more in detail what residues are acetylatilated in a ppGpp-dependent manner, and how does it impair lacZ transcription. But, as it is now discussed in the modified text (lines 317-318), this will be the focus of future manuscripts.

RC: Ok

9. Finally, regarding the conclusion that yfiQ expression is regulated by the levels of ppGpp (Fig. 3), the authors must indicate in the discussion section how the alarmone may exert its regulatory functions.

We absolutely agree with the reviewer. As previously said, genes directly stimulated by ppGpp has an A/T rich discriminator. As shown by Castaño-Cerezo et al. 2011, the expression of yfiQ is stimulated in stationary phase, but it does not respond to the alternative sigma factor RpoS, suggesting that any effect must be direct. This is now discussed at the modified version of the manuscript (lines 297-299).

RC: Ok

Minor comments.

1) Lines 98-101: Please describe in more detail this protocol, indicate the substrate used and specify how the Miller Units were determined.

This protocol is a well established standard in our field, but, as suggested by the reviewer, more information has been added.

RC: Comment appropriately addressed.

2) Lines 123-126. Please show as a supplementary Figure, a typical growth curve to see the diauxic effect and clearly indicate in this curve the OD600nm at which most of the sample were taken for biochemical and immunological determinations.

We agree that adding a diauxic growth curve will help to understand some parts of the manuscript; it is now a panel of figure S5. However, we believe that indicating there the OD at which most of the samples were taken will be misleading for the reader, considering that only the experiment shown in figure S5 was made during diauxic shift. Instead, we have stressed in all the figure legends that the samples were taken during exponential phase (at an OD600nm of 0.1).

RC: Comment appropriately addressed.

3) Figs. S3 and S7. Please describe in the Figure Legends the purpose of the Western Blotting of RpoA in these experiments.

This is now added to the figure legends as sugg

7. PLOS authors have the option to publish the peer review history of their article (what does this mean?). If published, this will include your full peer review and any attached files.

Reviewer #1: No

Reviewer #2: No

---

## [Editor Report · Acceptance letter]

19 Oct 2021

PONE-D-21-19172R1 

The secondary messenger ppGpp interferes with cAMP-CRP regulon by promoting CRP acetylation in *Escherichia coli*. 

Dear Dr. Fernández-Coll:

I'm pleased to inform you that your manuscript has been deemed suitable for publication in PLOS ONE. Congratulations! Your manuscript is now with our production department. 

Kind regards, 

on behalf of

Dr. Benjamin J. Koestler 

Academic Editor

PLOS ONE